# Biodiversity of Calanoida Copepoda in Different Habitats of the North-Western Red Sea (Hurghada Shelf)

**Hamdy Abo-Taleb** [1] , **Mohamed Ashour** [2,*] , **Ahmed El-Shafei** [3,4,*] , **Abed Alataway** [3] and **Mahmoud M. Maaty** [5,*]

1   Zoology Department, Faculty of Science, Al-Azhar University, Cairo, Nasr City 11823, Egypt; hamdy.ali.hamdy@gmail.com
2   Invertebrate Aquaculture Lab., National Institute of Oceanography and Fisheries (NIOF), Alexandria Branch 21556, Egypt
3   Prince Sultan Bin Abdulaziz International Prize for Water Chair, Prince Sultan Institute for Environmental, Water and Desert Research, King Saud University, Riyadh 11451, Saudi Arabia; aalataway@ksu.edu.sa
4   Department of Agricultural Engineering, College of Food and Agricultural Sciences, King Saud University, Riyadh 11451, Saudi Arabia
5   Hydrobiology Lab., Marine Environment Division, National Institute of Oceanography and Fisheries (NIOF), Hurghada Branch 84712, Egypt
*   Correspondence: microalgae_egypt@yahoo.com, (M.A.); aelshafei1bn.c@ksu.edu.sa (A.E.-S.); mahmoudmaaty1@yahoo.com (M.M.M.); Tel.: +966-11-4678504 (A.E.-S.)

**Abstract:** Little is known about the diversity of Calanoida, Copepoda, in different habitats of the north-western Red Sea. In this study, biodiversity of Calanoida, Copepoda, during the cold and warm seasons of 2017, were observed at 12 stations belonging to four different habitats (coral reef (CR), sheltered shallow lagoons (SSL), seagrass (SG), and open deep-water (ODW) habitats) in the Hurghada shelf, north-western Red Sea. SSL habitats were the most affected by environmental conditions, especially temperature, salinity, and depth. Some calanoid species were restricted to certain habitats, others were adapted to live in more than one habitat, while some species showed a wide distribution in all habitats. ODW habitats showed maximum diversity and density of the calanoid species. The effects of temperature and salinity were distinct in the SG and SSL. The results clearly showed that different Red Sea habitats affected the biodiversity of calanoid copepods.

**Keywords:** calanoida; copepoda; biodiversity; Red Sea; Coral reef; seagrass; habitats

## 1. Introduction

The coast of Hurghada has unique marine habitats. The waterfront region of Hurghada is described as coral reefs stretching out parallel to the shoreline, and is amongst the most important touristic territories influencing national revenue. Moreover, beside coral reef habitats, the Hurghada shelf has different prevailing marine habitats, such as sheltered shallow lagoons, seagrass, open deep-water habitats, and mangroves. Thus, it is distinct amongst the most appealing resorts in Egypt. The territory includes several human activities that are influenced by ecological changes [1]. Among the different habitats and ecosystems, an enormous number of species are found to be associated with a single and/or multiple habitats [2].

Marine zooplankton, especially the Copepoda, are considered an important bio-indicator of environmental changes associated with global warming. Copepoda represent more than 50% of planktonic metazoan biomass in the marine environment and are considered a secondary producer in

marine food webs [2–5]. Furthermore, the natural productivity of planktonic organisms, especially Copepoda, is very important for the environmental assessment of aquaculture potential [6]. Calanoida Copepoda are considered a dominant component of zooplankton in most marine water bodies worldwide; they are noteworthy because they are relatively the most abundant and diverse among Copepoda orders, under certain conditions [7]. Calanoida play an important role as energy transporters to the higher trophic levels in the marine ecosystem. This is because they consume the most amount of the primary production [8], owing to their role in the transport of organic matter to deep layers (either through faeces and their bodies when they die, or through daily vertical migration), and in the 'biological' pump [9,10]. Moreover, Calanoida represent major preys of mesopelagic and bathypelagic fish [11]. Chiffings [12] noted that among the 46 groups of zooplankton in the Northern Red Sea, and 60 groups in the southern part, Calanoida Copepoda were the most important. Little is known about their diversity in the surrounding waters of the different Red Sea habitats. Delalo [13], presenting the first study on the Red Sea zooplankton, implicitly included Calanoida, whilst Halim [14] introduced a review of earlier studies of zooplankton, including Calanoida, followed by further studies by Weikert [15,16]. Schmidt [17] was the first to study the abundance and diversity of Copepoda in the Red Sea at one site in the Gulf of Aqaba. The first specialized detailed study on Calanoida in the Red Sea was presented by Almeida Prado-Por and Por [18], followed by studies by Almeida Prado-Por [19–21] on the temporal and spatial distributions of the same group. No recent studies have given particular attention to the study of the ecology and biodiversity of Calanoida in different habitats of the Red Sea.

The current study is restricted to certain habitats in the Red Sea, and focuses on the relationship between the components of the ecosystem and Calanoida diversity and its effects. Literature describing the effect of habitat types on Calanoida Copepoda distribution is also sparse or even lacking. The current study aims to investigate the composition, abundance, and distribution structure, including habitat preference of Calanoida Copepoda species, in relation to different marine habitats in the cold and warm seasons in the Hurghada waterfront region, which is one of the most important tourist cities in Egypt with a large number of human activities in the north-western Red Sea. In addition, the most important environmental parameters affecting the calanoid distribution structure were investigated.

## 2. Materials and Methods

### 2.1. Study Area

All sampling stations (Figure 1) were in the Hurghada shelf located in the north-western part of the Egyptian coast of the Red Sea (27.30° and 27.326° N and 33.788° and 33.803° E). Twelve stations were selected to represent four different marine habitats: Coral Reef (CR), Seagrass (SG), Sheltered Shallow Lagoons (SSL), and Open Deep-Water (ODW).

Each habitat had three replicates and each sampling station was determined using a handheld Global Positioning System. Stations CR1, CR2, and CR3 were defined as coral reef habitats (depth 9–12 m). CR1 and CR2 were adjacent to the Abu-Sadaf reef, while station CR3 was located at the outer margin of the Abu Galawa Island adjacent to the reef edge and represented the coral reef zone. Stations SG1, SG2, and SG3 (depth 5–6.5 m) were defined as seagrass habitats and were located around the National Institute of Oceanography and Fisheries (NIOF), Hurghada Branch. Stations SSL1, SSL2, and SSL3 were defined as sheltered shallow lagoons habitats (depth 2–5 m). Stations ODW1, ODW2, and ODW3 were defined as open deep-water habitats (depth 29–56 m). The location and stations are shown in Figure 1.

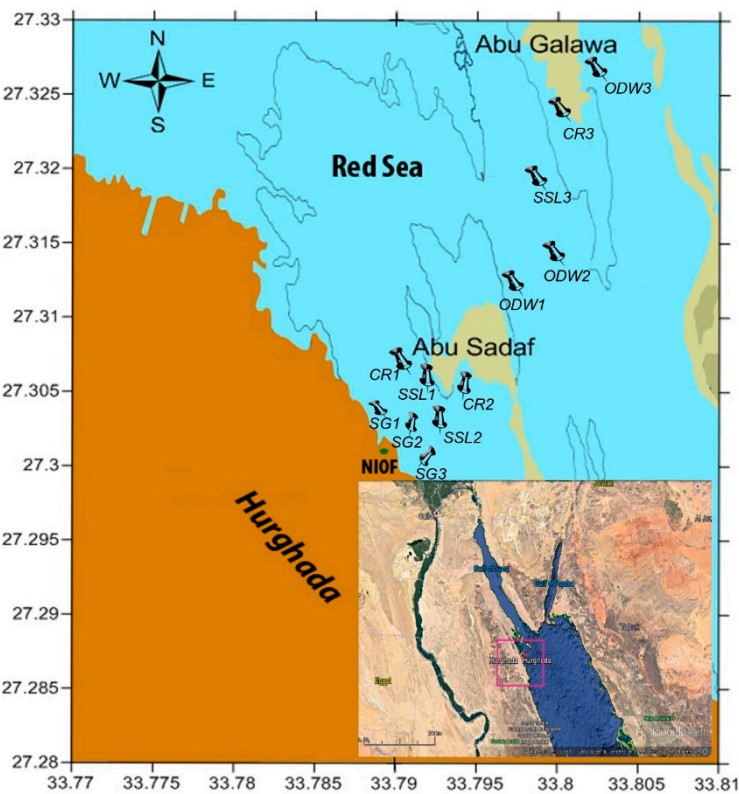

**Figure 1.** Map illustrating the location of sampling stations in the Red Sea along the coast of Hurghada, Egypt.

*2.2. Sampling and Analysis*

Samples were collected during the cold season (February, winter 2017) and warm season (August, summer 2017) between 06:18 and 09:39 (local time), during daylight hours. The most prevailing water physico-chemical parameters (temperature, salinity, pH and dissolved oxygen (DO) were measured. Temperature (°C) was measured with an ordinary thermometer graduated to 0.1 °C attached to the water sampler. Salinity was determined using an optical refractometer. The pH values were detected using a digital pH meter (model 201/digital pH meter). Dissolved oxygen (mg/l) was determined according to the Winkler method [22].

Calanoida Copepoda samples were collected as per protocols described by Abo-Taleb and Gharib [2]. Samples were collected using a standard plankton net (No. 25) of 55 μm mesh size with a diameter of 55 cm at the mouth. The net was lowered from the boat and towed horizontally along the surface layer of each station for 10 min with a towing speed of 2 knots, avoiding the reef edges. The net was equipped with a digital flow meter to calculate the efficiency of the filtration. After each tow, the net was rinsed thoroughly by dipping in seawater. This operation was performed thrice from three different locations at each station (N = 9 for each station, N = 27 for each habitat, N = 108 for each season). After sample collection, samples were transferred immediately to a small glass bottle and preserved in 4% neutralized formalin, and then, sample volume was adjusted to 100 mL. Each sample was examined in a Petri dish under a stereomicroscope and non-copepod groups were removed. The average of copepods was counted and used to estimate the copepod abundance, which is expressed as individual/m$^3$. For estimation of standing crop, sub-samples of 5 mL were transferred to a counting chamber using a plunger pipette (three subsamples from each sample were analyzed and the average was obtained). Standing crop was calculated and estimated as individuals per cubic meter as described by Santhanam and Srinivasan [23] using the following formula:

$$N = \frac{n \cdot v}{V \cdot S} \tag{1}$$

$$V = \pi\, r^2\, d\, f \qquad (2)$$

where *N*: Total number of zooplankton per cubic meter, *n***:** Average number of zooplankton in 5 mL of the sample, *v*: Volume of concentrated sample (100 mL), *V*: Volume of total filtered water (m³), *S*: volume of subsamples (5 mL), $\pi\, r^2$**:** The total area of the net mouth. *d*: Length of the tow by the net, *f*: Filtration coefficient (calculated by dividing the reading of the flow meter attached to the plankton net by the reading obtained without a bucket; as the value of filtration coefficient approaches 1, the net has high filtration efficiency).

### 2.3. Species Identification and Dissection

Species dissection and identification was performed using a stereomicroscope. The accurate identification of Calanoida Copepoda was performed by dissecting each species using fine needles on a glass slide and using a mixture of water, glycerine, and alcohol in the proportion 2:1:1. Only adults have been identified, while the immature stages have been counted altogether. The following references were consulted for identification: Sars [24], Newell and Newell [25], Rose [26], Mori [27], González and Bowman [28], Silas and Pillai [29], Bradford-Grieve [30], Boltovskoy et al. [31], and Conway et al. [32].

### 2.4. Statistical Analysis

Multiple correlation analysis and multiple regression analysis were computed using MINITAB Release 16, depending on fauna (abundance and diversity) and the measured ecological parameters with a significance level of $p \leq 0.05$. PcOrd 5.0 (functions of data analysis to analyze various environmental and ecological data within the framework of the Euclidean exploratory methods) was applied to detect correlations between the different environmental parameters measured at the twelve stations representing the four prevailing Red Sea habitats. In addition, PcOrd 5.0 was used for (1) the hierarchical clustering, and two ways hierarchical clustering applied for the stands, depending on their zooplankton species composition using Euclidean similarity matrix; and (2) the distribution of stations based on the zooplankton species composition in each one versus the environmental condition using PCA methods.

## 3. Results

### 3.1. Physico-Chemical Parameters

The investigated physico-chemical parameters (Table 1) showed that temperature ranged from 28.00 to 36.50 °C, salinity varied from 35.20 to 44.00 ppt, and dissolved oxygen (DO) ranged from 8.08 to 5.50 mg/l in the cold and warm seasons, respectively. The pH in all observed stations in both seasons were alkaline (9.00 to 7.80) except in the SG stations, which were neutralized and tended to be slightly alkaline (7.00 to 7.70). Among the different surveyed habitats, the maximum temperature and salinity was observed at the SSL habitats during the warm season (34.35 ± 1.99 °C and 42.93 ± 0.95 ppt, respectively). The SSL habitats also recorded the minimum temperature and salinity (28.13 ± 0.15 °C and 38.13 ± 2.57 ppt, respectively) during the cold season.

**Table 1.** Studied abiotic factors in different habitat and stations during cold (C) and warm (W) seasons.

| Habitat Types | Stations | Depth (m) | Temperature °C | | Salinity g/l (ppt) | | DO mg/l | | pH | |
|---|---|---|---|---|---|---|---|---|---|---|
| | | | C | W | C | W | C | W | C | W |
| Seagrass (SG) | SG1 | 6 | 29.10 | 30.90 | 41.03 | 41.61 | 7.68 | 6.22 | 7.70 | 7.40 |
| | SG2 | 5 | 29.50 | 32.50 | 41.20 | 41.00 | 7.62 | 6.25 | 7.30 | 7.50 |
| | SG3 | 6.5 | 29.00 | 31.00 | 41.30 | 42.00 | 7.40 | 6.00 | 7.50 | 7.00 |
| | Av ± SD | 5.83 ± 0.76 | 29.20 ± 0.26 | 31.47 ± 0.9 | 41.18 ± 0.14 | 41.54 ± 0.50 | 7.57 ± 0.15 | 6.16 ± 0.14 | 7.5 ± 0.20 | 7.30 ± 0.26 |
| Coral Reef (CR) | CR1 | 12 | 28.70 | 31.00 | 41.45 | 41.78 | 7.29 | 6.15 | 8.10 | 8.10 |
| | CR2 | 9 | 28.50 | 30.00 | 41.35 | 41.88 | 7.34 | 6.16 | 8.40 | 8.30 |
| | CR3 | 13 | 28.40 | 29.90 | 41.11 | 41.76 | 7.45 | 6.24 | 8.00 | 8.20 |
| | Av± SD | 11.33 ± 2.08 | 28.53 ± 0.15 | 30.30 ± 0.61 | 41.30 ± 0.17 | 41.81 ± 0.06 | 7.36 ± 0.08 | 6.18 ± 0.05 | 8.17 ± 0.21 | 8.20 ± 0.10 |
| Sheltered Shallow Lagoon (SSL) | SSL1 | 2.8 | 28.00 | 34.00 | 39.20 | 42.60 | 7.23 | 6.13 | 7.90 | 8.40 |
| | SSL2 | 2 | 28.10 | 36.50 | 35.20 | 44.00 | 6.90 | 5.50 | 8.00 | 9.00 |
| | SSL3 | 5 | 28.30 | 32.56 | 40.00 | 42.20 | 7.24 | 6.13 | 8.00 | 8.40 |
| | Av± SD | 3.27 ± 1.55 | 28.13 ± 0.15 | 34.35 ± 1.99 | 38.13 ± 2.57 | 42.93 ± 0.95 | 7.12 ± 0.19 | 5.92 ± 0.36 | 7.97 ± 0.06 | 8.60 ± 0.35 |
| Open Deep Water (ODW) | ODW1 | 29 | 28.50 | 30.10 | 41.01 | 41.23 | 7.89 | 6.25 | 8.10 | 8.20 |
| | ODW2 | 31 | 29.00 | 29.80 | 41.07 | 40.32 | 7.59 | 6.32 | 7.80 | 8.00 |
| | ODW3 | 56 | 28.00 | 30.20 | 41.12 | 40.22 | 8.05 | 6.85 | 7.90 | 8.00 |
| | Av± SD | 38.67 ± 15.04 | 28.50 ± 0.50 | 30.03 ± 0.21 | 41.07 ± 0.06 | 40.59 ± 0.56 | 7.84 ± 0.23 | 6.47 ± 0.33 | 7.93 ± 0.15 | 8.07 ± 0.12 |

Represented Data are mean ± SD. Different superscript letters in each column indicate significant differences ($p < 0.05$). Statistical differences ($p < 0.05$, n = 3) between groups are indicated by different letters.

In the warm season, among stations, station SSL2 recorded the highest temperature (36.50 °C), salinity (44.00 ppt), and pH (9.00), while it recorded the lowest DO (5.50 mg/l). Moreover, station SSL2 had the lowest depth (2 m) of all investigated stations. ODW habitats recorded the maximum average DO, both in cold (7.84 ± 0.23 mg/l) and warm (6.47 ± 0.33 mg/l) seasons. The highest DO values in the cold (8.05 mg/l) and warm (6.85 mg/l) seasons were recorded at station ODW3, considered the deepest site (56 m). In contrast, SSL stations recorded the minimum average of DO, both in cold (7.12 ± 0.19 mg/l) and warm (5.92 ± 0.36 mg/l) seasons.

## 3.2. Dynamics of Species in Different Habitats

Calanoida Copepoda were represented by 38 species belonging to 18 genera and 13 families, in addition to the immature forms (nauplius larvae and copepodite stages), distributed in all the studied Red Sea habitats (Table 2). Six species were exclusive of the warm period, while four species were exclusive of the cold period (Figures 2 and 3, respectively). During the warm season, among all habitats, the highest diversity was observed in ODW1 and ODW3 (17 and 16 species, respectively) and abundance (1400 and 1470 org./m$^3$, respectively), while station ODW3 sustained the highest diversity (27 species) and abundance (387 org./m$^3$) during the cold season. The lowest diversities and abundances were recorded in the SG and SSL habitats during the cold and warm seasons, respectively.

**Table 2.** List of families and species of calanoid copepods which identified in the present study.

| **Acartiidae** | | **Pontellidae** |
|---|---|---|
| *Acartia bispinosa* Carl 1907 | | *Calanopia elliptica* (Dana 1846; 1849) |
| *Acartia clausi* Giesbrecht 1889 | **Clausocalanoidae** | *Labidocera detruncata* (Dana 1849) |
| *Acartia danae* Giesbrecht 1889 | *Clausocalanus arcuicornis* (Dana 1849) | *Labidocera euchaeta* (Dana 1846) |
| *Acartia erythraea* Giesbrecht 1889 | *Clausocalanus furcatus* (Brady 1883) | *Labidocera madura* Scott A.1909 |
| *Acartia fossae* Gurney 1927 | *Clausocalanus pergens* Farran 1926 | *Labidocera pavo* Giesbrecht 1889 |
| *Acartia negligens* Dana 184 | | *Pontellina plumata* (Dana 1849) |
| **Calanidae** | **Euchaetidae** | **Pseudodiaptomidae** |
| *Nannocalanus minor* (Claus 1863) | *Euchaeta indica* Wolfenden 1905 | *Pseudodiaptomus hessei* (Mrázek 1894) |
| | *Euchaeta marina* (Prestandrea 1833) | |
| **Candaciidae** | **Paracalanidae** | **Scolecitrichidae** |
| *Candacia bradyi* Scott A. 1902 | *Acrocalanus gibber* Giesbrecht 1888 | *Scolecithrix danae* (Lubbock 1856) |
| *Candacia longimana* (Claus 1863) | *Calocalanus pavo* (Dana 1852) | **Temoridae** |
| *Candacia simplex* (Giesbrecht 1889) | *Mecynocera clausi* Thompson I.C. 1888 | *Temora discaudata* Giesbrecht 1889 |
| *Candacia truncata* (Dana 1849) | *Paracalanus aculeatus* Giesbrecht 1888 | *Temora stylifera* (Dana 1849) |
| | | *Temora turbinata* (Dana 1849) |
| **Centropagoidae** | *Paracalanus parvus* (Claus 1863) | |
| *Centropages elongatus* Giesbrecht 1896 | **Phaennidae** | **Tortanidae** |
| *Centropages gracilis* (Dana 1849) | *Phaenna spinifera* Claus 1863 | *Tortanus recticauda* (Giesbrecht, 1889) |
| *Centropages orsini* Giesbrecht 1889 | | |
| *Centropages violaceus* (Claus 1863) | | |

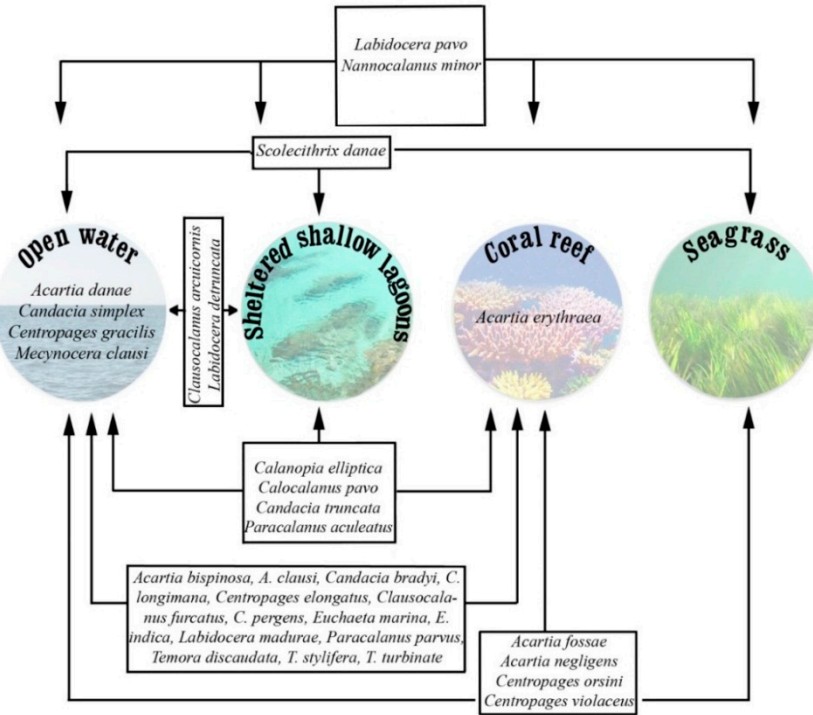

**Figure 2.** Schematic showing the distribution of different Calanoida species in different habitats during the cold season (arrows indicate the habitat in which the organisms were found).

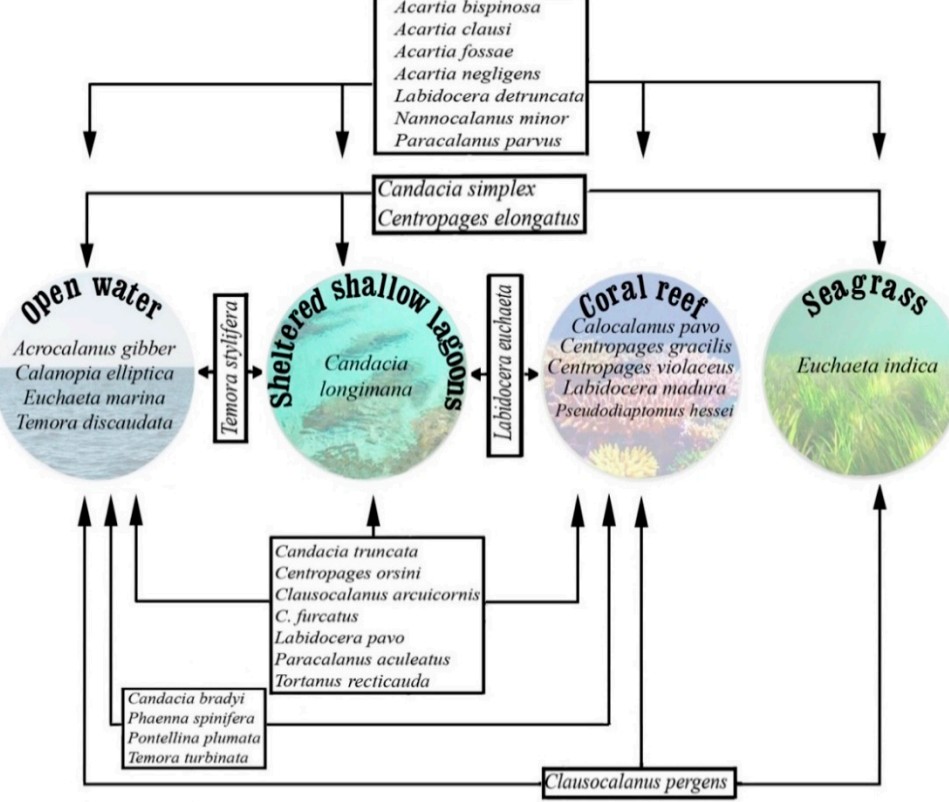

**Figure 3.** Schematic showing the distribution of different calanoid species in different habitats during the warm season (arrows indicate the habitat in which the organisms were found).

Figure 2 shows the distribution of different Calanoida species at different habitats during the cold season. Some species were restricted to certain habitats; *Acartia danae, Candacia simplex, Centropages gracilis*, and *Mecynocera clausi* were restricted to ODW; *Acartia erythraea* characterized the CR habitat; while no species were found to be restricted to the SG or SSL habitats during this season (Figure 2). *Labidocera pavo* and *Nannocalanus minor* were the only two species that were widespread in all investigated habitats. In contrast, some species were found in ODW, CR, and SG habitats, while others were recorded in ODW, CR, and SSL habitats (Figure 2). *Scolecithrix danae* was the only species having the ability to inhabit SG, ODW, and SSL habitats. *Clausocalanus arcuicornis* and *Labidocera detruncata* were recorded only in ODW and SSL habitats. There was no species that occurred both in the CR and SSL habitats. The highest number of species (14 species) were found in ODW and CR (Figure 2).

Figure 3 shows the distribution of Calanoida species in different habitats during the warm season. Some species were restricted to certain habitats, such as *Candacia longimana* at the SSL habitats, *Euchaeta indica* at the SG habitats, and *Acrocalanus gibber, Calanopia elliptica, Euchaeta marina*, and *Temora discaudata* at the ODW habitats. CR habitats were characterized by five species that were not recorded in any other habitat during this season. Seven species had a wide distribution and were recorded in all habitats (Figure 3).

In contrast, some species adapted to survive in different habitats and were found in ODW, SSL, and CR habitats. *Clausocalanus pergens* was also observed in ODW, CR, and SG habitats. *Candacia simplex* and *Centropages elongatus* were observed in ODW, SSL, and SG habitats. Finally, some species were found to be restricted to two specific habitats; *Candacia bradi, Phaenna spinifera, Pontellina plumata*, and *Temora turbinata* were constrained to ODW and CR habitats; *Temora stylifera* was recorded in only ODW and SSL habitats; and *Labidocera euchaeta* in only SSL and CR habitats (Figure 3).

Generally, *Labidocera,* as a genus, and *Nannocalanus minor* could be found in all habitats during different seasons. In contrast, *Centropages gracilis* was recorded in only one habitat; however, this habitat changed from one season to the other. It was a CR inhabitant during the warm season but during the cold season, they were found only in ODW habitats. Acartiidae, which was widespread during the warm season, demonstrated a different trend in the cold season. It became restricted to only ODW (*A. danae*) or CR (*A. erythraea*) or both habitats together (*A. bispinosa and A. clausi*), while some species could survive in the two previous habitats in addition to the SG habitat, such as *A. fossae* and *A. negligens*. In contrast, during the cold season, there was no species of this genus restricted to SG habitats alone.

Regardless of the habitat types, results of the spatial distribution of species within all studied stations (Figure 4) showed that during the cold season, the highest occurrence (92%) was recorded for two species (*N. minor* and *L. pavo*). *A. erythraea* and *M. clausi* occurred with a value of 8% in one station during this season. There were no species having a wide distribution over all the studied stations, except the immature forms. In contrast, during the warm season, three species were recorded at all stations with 100% frequency of occurrence. In contrast, nine species were restricted to a single station, with an occurrence value of 8% of the studied sites (Figure 4).

The differences in abundance of common species at different habitats were analyzed to see whether the difference was significant. The results showed that during the cold season, there were no significant differences ($p < 0.05$) observed in abundances of the common species (*L. pavo* and *N. minor*) in different habitats, as shown in Table 3. In contrast, during the warm season, there were no significant differences ($p < 0.05$) observed in abundances of the common species at different habitats. The only highest significant difference ($p < 0.05$) was observed with *N. minor* in the ODW habitat, followed by SSL, SG, and CR, as shown in Table 3.

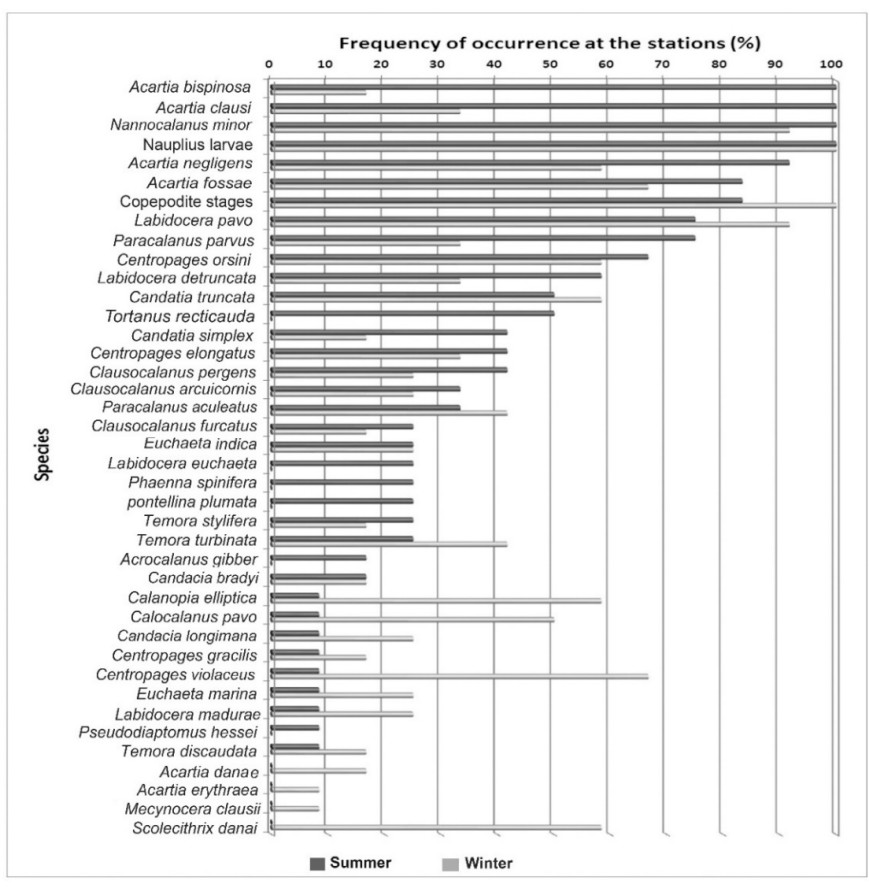

**Figure 4.** Histogram showing the frequency of occurrence of Calanoida Copepoda at different stations during the investigation period.

**Table 3.** Diversity and density of Calanoida species, related to different stations and habitats.

| Habitat types | Stations | Cold Season | | Warm Season | |
|---|---|---|---|---|---|
| | | Diversity sp. No. | Density Org./m³ | Diversity sp. No. | Density Org./m³ |
| Seagrass (SG) | SG1 | 7 | 111 | 11 | 889 |
| | SG2 | 7 | 148 | 11 | 963 |
| | SG3 | 7 | 112 | 11 | 892 |
| | Total | 7 | 124 | 11 | 915 |
| Coral Reef (CR) | CR1 | 7 | 127 | 14 | 1233 |
| | CR2 | 10 | 209 | 12 | 1387 |
| | CR3 | 20 | 367 | 15 | 1276 |
| | Total | 25 | 234 | 24 | 1299 |
| Sheltered Shallow Lagoon (SSL) | SSL1 | 6 | 157 | 12 | 1099 |
| | SSL2 | 8 | 145 | 13 | 1063 |
| | SSL3 | 6 | 338 | 14 | 689 |
| | Total | 9 | 213 | 19 | 950 |
| Open Deep Water (ODW) | ODW1 | 19 | 316 | 17 | 1400 |
| | ODW2 | 17 | 236 | 13 | 1065 |
| | ODW3 | 27 | 387 | 16 | 1470 |
| | Total | 31 | 313 | 26 | 1312 |
| Total | | 32 | 221 | 34 | 1119 |

### 3.3. Seasonal Variation

The diversity and density of Calanoida species in relation to different stations and habitat types (SG, CR, SSL, and ODW) are shows at Table 4. The species diversity and abundance varied in relation to seasons and habitat types. Generally, regardless of the habitat types, higher diversity, and abundance (34 species and 1119 org./m$^3$, respectively) was observed in the warm season rather than the cold season (32 species and 221 organisms/m$^3$, respectively), Table 4. Regardless of the seasons, ODW habitats had the highest diversity and average abundance during the warm (26 species and 1312 org./m$^3$, respectively) and cold seasons (31 species and 313 org./m$^3$, respectively), while SG habitats recorded the lowest diversity and average abundance during the warm (11 species and 915 org./m$^3$, respectively) and cold seasons (7 species and 124 org./m$^3$, respectively), Table 4.

**Table 4.** Abundances of the common species in different habitats in cold and warm seasons.

| Habitats | Cold Season | | | | | Warm Season | | | |
|---|---|---|---|---|---|---|---|---|---|
| | *Labidocera pavo* | *Nannocal-anus minor* | *Acartia bispinosa* | *Acartia clausi* | *Acartia fossae* | *Acartia negligens* | *Labidocera detruncata* | *Nannocal-anus minor* | *Paracalanus parvus* |
| SG | 8.33 ± 4.50$^a$ | 4.00 ± 0.00$^a$ | 13.3 ± 1.5$^a$ | 17.3 ± 1.5$^a$ | 13.0 ± 2.0$^a$ | 2.0 ± 1.0$^a$ | 1.3 ± 0.6$^a$ | 22.0 ± 3.0$^b$ | 4.3 ± 1.5$^a$ |
| CR | 4.00 ± 4.00$^a$ | 5.33 ± 2.30$^a$ | 18.3 ± 2.3$^a$ | 18.3 ± 24.8$^a$ | 11.3 ± 16.3$^a$ | 4.3 ± 3.5$^a$ | 1.3 ± 2.3$^a$ | 18.3 ± 2.3$^b$ | 0.7 ± 0.6$^a$ |
| SSL | 10.0 ± 10.14$^a$ | 3.00 ± 4.35$^a$ | 33.3 ± 25.5$^a$ | 13.7 ± 8.7$^a$ | 11.7 ± 11.1$^a$ | 2.3 ± 1.5$^a$ | 1.7 ± 2.1$^a$ | 26.7 ± 14.6$^b$ | 0.3 ± 0.6$^a$ |
| ODW | 11.33 ± 6.65$^a$ | 7.33 ± 8.50$^a$ | 28. ± 8.9$^a$ | 22.7 ± 10.6$^a$ | 5.3 ± 2.3$^a$ | 1.7 ± 2.1$^a$ | 1.3 ± 2.3$^a$ | 52.0 ± 8.9$^a$ | 10.7 ± 16.7$^a$ |
| mean ± SD | 8.42 ± 6.45$^a$ | 4.92 ± 4.52 | 23.3 ± 14.2 | 18 ± 12.6 | 10.3 ± 9.0 | 2.6 ± 2.2 | 1.4 ± 1.7 | 29.8 ± 15.7 | 4.0 ± 8.4 |

Represented Data are mean ± SD. Different superscript letters in each column indicate significant differences ($p < 0.05$). Statistical differences ($p < 0.05$, n = 3) between groups are indicated by different letters.

The immature stages (nauplius and copepodites) showed an obvious variation between seasons, as well as between different habitats. The winter showed relatively limited abundance (average 163.3 ± 145.8 individuals/m$^3$) compared to the huge increase that occurred during the summer (average 895.1 ± 326.5 individuals/m$^3$). During the cold season, the highest abundance of immature forms was recorded in open water habitats (mean 289.5 ± 221.04 individuals/m$^3$), while the lowest numbers were obtained from seagrass habitats (mean 79 ± 31.2 individuals/m$^3$). This case was relatively reverse during the warm season, where the lowest abundance of immature stages was noted from open water habitats (762.5 ± 534.59 individuals/m$^3$), while the largest numbers ever during this season were in reef habitats (1169.5 ± 662.96 individuals/m$^3$), Table 5.

**Table 5.** Abundance (range, average ± SD) of the immature stages of Copepoda (individuals/m3) at different habitats and seasons.

| Seasons | Habitats | Range | Average ± SD |
|---|---|---|---|
| Cold season | SG | 76–82 | 79 ± 31.2 |
| | CR | 72–140 | 106 ± 52.73 |
| | SSL | 73–284 | 178.5 ± 119.63 |
| | ODW | 101–478 | 289.5 ± 221.04 |
| Warm season | SG | 705–837 | 771 ± 439.02 |
| | CR | 1047–1292 | 1169.5 ± 662.96 |
| | SSL | 589–1166 | 877.5 ± 549.69 |
| | ODW | 360–1165 | 762.5 ± 534.59 |

### 3.4. Multiple correlation analysis

Multiple correlation analyses between Calanoida abundance and diversity and the prevailing environmental conditions (temperature, salinity, DO, and pH) were performed with $p \leq 0.05$ as the significance level. During the cold season, abundance of *Acartia erythraea* was positively correlated with pH (r = 0.753), while abundance of *Candacia simplex*, *Centropages violaceus*, and *Temora stylifera* were negatively correlated with temperature (r = −0.633, −0.649, and −0.637, respectively). The effect of water depth was evident in the abundance of some species, which decreased in sheltered shallow

lagoons habitats and gradually increased with the water depth, recording a positive correlation between their abundance and water depth (r = 0.845, *Candacia simplex*; r = 0.732, *Centropages orsinii*; r = 0.630, *Clausocalanus pergens*; and r = 0.749, *Labidocera madurae*). DO was positively correlated with abundance of *C. simplex* (r = 0.842), *C. orsinii* (r = 0.875), and *L. madurae* (r = 0.615), while salinity was positively correlated with *Labidocera detruncata* abundance (r = 0.670) and negatively with that of *Centropages orsinii* (r = −0.725). Calanoida diversity during the cold season was positively correlated with water depth (r = 0.794) and DO (r = 0.694), while negatively correlated with temperature (r = −0.655) and salinity (r = −0.724). In contrast, Calanoida abundance was positively correlated with the abundance of *Acartia fossae* (r = 0.693), *Centropages elongatus* (r = 0.663), and *Paracalanus parvus* (r = 0.758).

During the warm season, abundance of two species, *Euchaeta indica,* and *Candacia simplex*, was negatively correlated with pH (r = −0.865 and −0.774, respectively). *Paracalanus aculeatus* abundance correlated positively with temperature (r = 0.617). Moreover, abundance of *Temora discaudata* was positively correlated with water depth (r = 0.717) and DO (r = 0.833). In contrast, total abundance of Calanoida in the studied habitats during the warm season was mainly controlled by the abundance of three species, *Acartia bispinosa, Acrocalanus gibber, and Tortanus recticauda*, which were correlated with the total Calanoida abundance (r = 0.617, 0.595, and 0.741, respectively).

Overall, during the warm season, the water depth was correlated positively with the dissolved oxygen (r = 0.929) and negatively with salinity (r= −0.885), while in the cold season the salinity correlated positively with depth (r = 0.853) and temperature (r= 0.613). There was a positive correlation between temperature and salinity (r = 0.622) in the warm season and a negative correlation between salinity and DO during the warm (r = −0.755) and cold (r = −0.805) seasons. This may explain the condition of environmental stress in the SSL habitats.

### 3.5. Multiple Regression Analysis

Multiple regression analysis of total Calanoida versus water depth, salinity, pH, temperature, and DO resulted in the following prediction regression equations:

$$\text{Abundance (Individuals/m}^3) = -677 + 1.56\,\text{Depth} + 7.3\,\text{Temp} + 13.2\,\text{Salinity} - 23.6\,\text{DO} + 21.9\,\text{pH} \quad (3)$$

$$\text{Diversity (Number of species)} = -125 + 0.180\,\text{Depth} - 0.54\,\text{Temp} + 3.32\,\text{Salinity} + 1.13\,\text{DO} + 0.76\,\text{pH} \quad (4)$$

Stepwise regression prediction equations: The prediction equations for estimating the abundance of the organisms resulted in some errors that could be minimised using the stepwise regression equations that were performed to exclude parameters, which were not strongly correlated (not significant, p > 0.05):

$$\text{Abundance (Individuals/m}^3) = 96.4 - 43\,\text{DO} + 1.42\,\text{Depth} \quad (5)$$

$$\text{Diversity (Number of species)} = -96.04 + 0.203\,\text{Depth} + 2.6\,\text{Salinity} \quad (6)$$

### 3.6. Cluster Analysis

Based on species composition, during the cold season, there was relative similarity between different stations and the maximum similarity was observed between SG habitats. There was high similarity (96.35%) during this season between the CR (CR1) and the SSL (SSL2) habitats. Among the cluster formed by all ODW and CR stations, CR3 formed a separate cluster that recorded minimum similarity (13.54) with the remaining stations, (Figure 5).

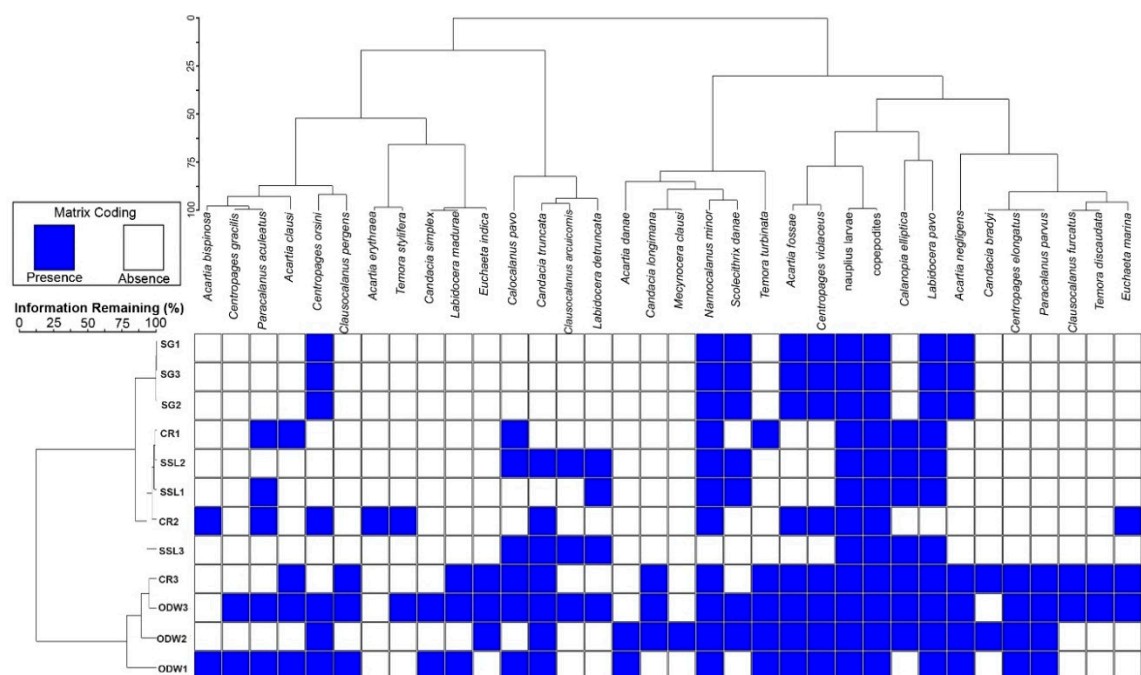

**Figure 5.** Dendrogram of two-way cluster analysis based on group average clustering linkage from Euclidean similarity matrix showing the similarity between studied stations during cold season.

In contrast, during the warm season, the highest similarity (99.62%) was recorded between SG stations (SG1 and SG3), followed by the similarity (93.59%) between SSL habitats (SSL1 and SSL2). The minimum similarity (7.32%) was recorded between SG and SSL habitats and ODW (ODW1) (significant difference, $p < 0.05$), (Figure 6).

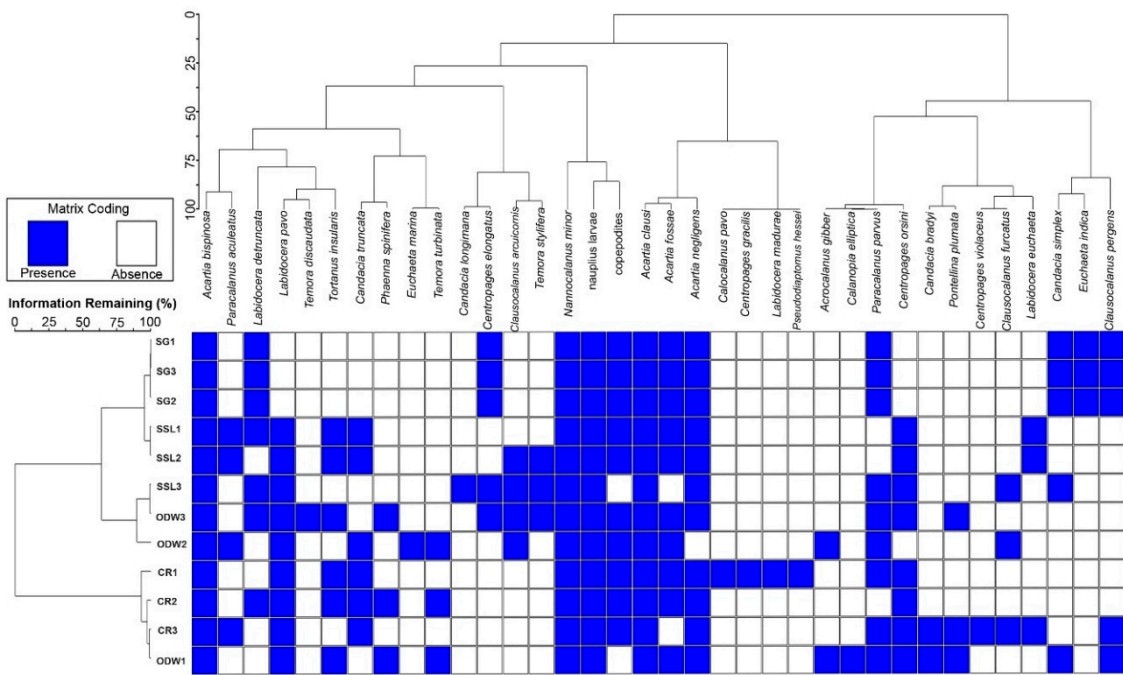

**Figure 6.** Dendrogram of two-way cluster analysis based on group average clustering linkage from Euclidean similarity matrix showing the similarity between studied stations during warm season.

### 3.7. Principal Component Analysis (PCA)

Five environmental conditions (descriptors) were used: temperature, salinity, DO, pH, and water depth. The relationship of these environmental variables with the first two axes of variation (component 1 and 2 with proportions of variance 18.9 % and 11%, respectively, during cold season, while 18.1%, and 11.5% for axis 1 and 2, respectively, during warm season) indicates the most effective and influential variables. PCA and the components of the environmental data biplot of the first two axes are shown in Figures 7 and 8 for cold and warm seasons, respectively. During the cold season, close correlation was observed between DO and depth contrary to the temperature along the first axis. These variables showed high DO values and depth at the ODW stations (ODW1 and ODW2) in contrast to the SG stations, CR1, and SSL that were more influenced by water temperature. The SSL (SSL1) was characterized by low salinity content (Figure 7).

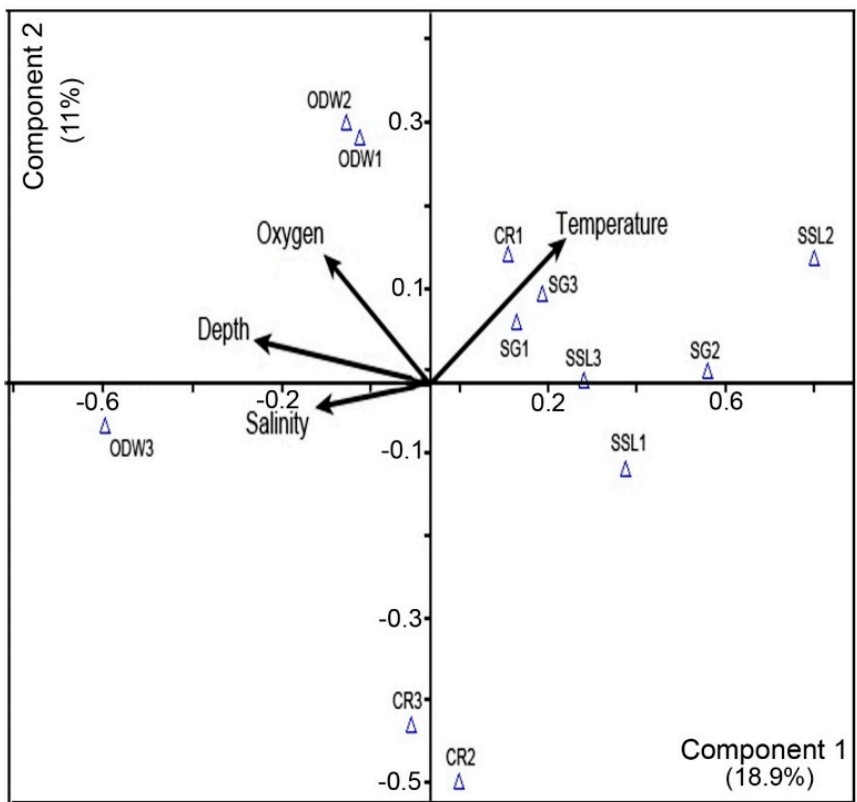

**Figure 7.** Principal Component Analysis (PCA) components projection on Axis 1 and Axis 2 showing the distribution of the stations based on the prevailing environmental condition during warm season.

In contrast, during the warm season, there were close correlations between temperature and salinity in contrast to the depth and DO along the first axis. The pH was obviously more correlated with the second axis. Variables associated with axis 1 (temperature and salinity) indicate an environmental stress condition gradient and clearly illustrates the tendency of SSL habitats being severely affected by environmental changes. Axis 2 is related to pH. PCA ordination of the data, according to the 12 reference stations, indicates the water quality differences between both the habitats and the sampling stations. Sample points were well distinguished according to each habitat. In ODW habitats, stations are clearly distinguished by being deeper, higher in DO content, and ecologically more stable. SG stations were characterized by low DO content and they were negatively correlated with pH values (Figure 8).

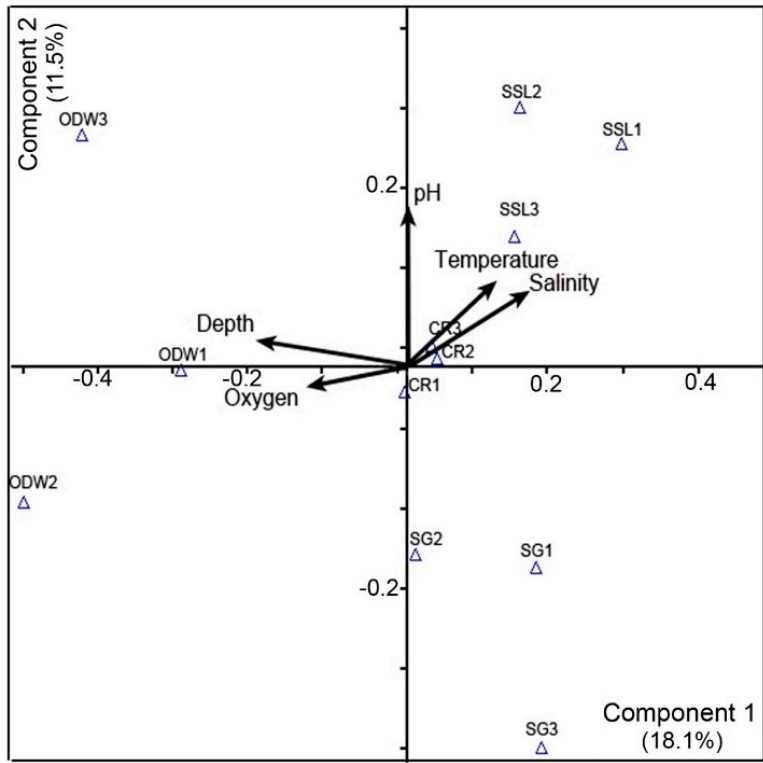

**Figure 8.** PCA components projection on Axis 1 and Axis 2 showing the distribution of the stations based on the prevailing environmental condition during cold season.

## 4. Discussion

The unique geographic location of the Red Sea makes the ecosystem highly sensitive to temperature and salinity, which is reflected in the distribution and diversity of plankton communities [20]. Plankton are considered very important bioindicator for the marine ecosystem and monitoring of the water quality. Many studies reported that the plankton (phyto and zoo) composition, distribution, diversity, and communities are affected by variations in environmental and nutritional conditions [2,6,33–43].

In the current study, the differences between the lowest (in cold season) and the highest (in warm season) temperature and salinity values were 8.5 °C and 8.8 ppt, respectively. SSL habitats were the most affected by environmental conditions, especially temperature, salinity, and depth. The alteration in temperature and salinity values clearly affected these habitats with minimum temperature and salinity values (28.1 °C and 35.2 ppt, respectively) during the cold season and the maximum values (36.5 °C and 44 ppt, respectively) during the warm season. These differences observed in the northern part of the Red Sea are clearly affecting the abundance and diversity of Calanoida Copepoda within the Red Sea, especially the SSL habitats [44]. PCA results indicated that the hydrological conditions, including temperature and salinity, have a great effect on the semi-enclosed areas such as sheltered shallow habitats. This may explain why there are no species restricted to such habitats during the winter [2].

Of all the studied habitats, the highest average temperature and salinity (34.4 °C and 42.9 ppt, respectively) were recorded during the warm season in shallow lagoon environments, which also showed the lowest values (28.1 °C and 38.1 ppt, respectively) during the cold season. This seasonal variation, and variability in other habitats of these two environmental factors, may be the main cause of the low diversity and abundance of Calanoida in these habitats. Many studies have indicated that there is a decline in Copepoda diversity because of changes in temperature and salinity [45]. These are considered the main factors that control and regulate the abundance and distribution of Copepoda, and of most other plankton organisms [46–53]. When excess food is available for aquatic organisms,

the species abundance will mainly depend on temperature because the rising temperature in an area with highly nutritional salts and food content causes eutrophication that can be a limiting factor to many Copepoda species [54–57].

Owing to the rapid response of Copepoda to environmental conditions and their short life cycle, they are considered a good bio-indicator for ecosystem status and health [58,59]. Copepoda are considered the dominant zooplankton group in many parts of the Red Sea; they represent 70% of the total metazooplankton numbers [60,61]. Calanoida are relatively the most dominant among copepod orders [7], especially in the Red Sea [61]. According to El-Serehy and Abdel-Rahman [62], Calanoida represented 64% of the total Copepoda species and comprised 47 species among a total of 74 Copepoda species recorded during their study, while Al-Najjar [61] recorded 34 Calanoida (62%) from a total of 55 Copepoda species, and they comprised 49.2% (38 species of a total 67 adult Copepoda) in the study by Abo-Taleb and Gharib [2]. Therefore, using these as bio-indicators is of great value; hence, the importance of this study.

In the present study, 38 species of Calanoida, in addition to the immature forms (nauplius larvae and copepodites), were distributed in different habitats of the Red Sea. Almeida Prado-Por [19] identified 31 Calanoida species from samples collected seasonally from a stratified tow from the Gulf of Aqaba in 1975; the author assumed that Calanoida plankton, both quantitatively and qualitatively, was poor. The present data indicated that although the total number of Calanoida species in winter (by a relatively marginal extent) was lower (32 species) than in summer (34 species), in the SSL and SG habitats that have a special nature, the species number in winter was always higher than in the summer, reflecting the effect of such habitats on the Calanoida community. The winter peak of Calanoida species numbers in the Red Sea has been previously noted by Almeida Prado-Por [19] and El-Serehy and Abdel-Rahman [62].

Calanoida recorded increases in densities during the warm season in all habitats. This noticeable summer peak of Calanoida in the Red Sea may follow the peak of the zooplankton in the Indian Ocean, that are introduced into the Red Sea through the Bab El-Mandeb Strait during spring, owing to the prevailing pattern of water exchange towards the Red Sea. In winter, the plankton abundance in the Red Sea reduces as the surface water becomes less energetic because of the deficiency of nourishment and lack of water exchange [44], explaining the significant decrease in densities during the cold season. Furthermore, Chen [63] and Li et al. [64] recorded that the reproduction frequency of Copepoda, hatching rate, and even mating recurrence increases when the temperature rises above 35 °C. Classification of the different habitats in the studied area of the Red Sea was based on the fact that every habitat is controlled by several factors, including environmental conditions, availability of food, natural enemies, and predators, in addition to the competition for the space. Gusmão and McKinnon [65] noted the effects of the presence of ecto- and endo-parasites, limitation of food, pollution, and ultraviolet radiation on Copepoda reproduction. These factors reflected on the distribution and abundance of the organisms and the consequences appear as the higher decrease in number of species and their densities in semi-enclosed and coral reef habitats than in the sandy open water. These results are consistent with those of El-Serehy and Abdel-Rahman [62] who observed higher decreases in the number of Copepoda species, as general, (total 48 species), specially Calanoida (35 species) in the coral reefs of the Red Sea at Hibika and Abu Galum than in the offshore sandy stations of Nuweiba and Ras Burka (74 Copepoda species involve 47 Calanoida), where 26 species of Copepoda were absent, including 12 species of the order Calanoida alone. Moreover, in the current study, the highest number of Calanoida species was recorded from open water habitats during the warm and cold seasons (26 and 31 species, respectively).

The results of the present study illustrated that the lowest species numbers and abundance during cold and warm seasons were recorded in seagrass habitats, followed by the shallow lagoons and the coral reef habitats, while the highest were recorded in the open deep-water areas. These results are consistent with those of Abo-Taleb and Gharib [2], who studied the distribution of epipelagic Copepoda in the same habitats as our study. Depending on habitat type, there is an inverse relationship

between Copepoda abundance and fish larvae. This finding was confirmed by Abu El-Regal [66] who studied fish larvae distribution and abundance along the coast of Hurghada. They noticed significant differences in larval abundance between different sites in the inshore areas; the highest abundance was recorded at the sheltered sites while the lowest was observed at open exposed sites.

In addition, Abo-Taleb and Gharib [2] studied the distribution of epipelagic Copepoda in the same habitats as our study and concluded that seagrass are the preferred habitats for the juveniles and fish larvae more than any other investigated habitat [67]. Many studies indicated that the stomach content of the seagrass and coral reef fishes and larvae comprised Calanoida as the bulk food items [2,62,68]. The location and abundance of fish larvae are strongly related to the habitat types and the adult spawning juveniles are known to utilise seagrass beds as a potential nursery habitat [67,69]. Fish larvae prefer seagrass substrates more than corals, for example, *Lethrinus* spp. dominantly inhabit seagrass beds in the juvenile life [68–70]. Hence, some offshore fish larvae seek seagrass habitats as nursery grounds [71]. Seagrass attract offshore larvae in high numbers owing to the presence of chemical odour cues of the plumes of some seagrasses that diffuse over a relatively wide distance [72].

Seagrass and coral reef habitats are considered a great attraction for many organisms from different trophic levels (more than 35 families of coral reef fish were recorded by El-Regal et al. [5] in the same area as the present study); this causes competition for food and space as their maximum levels are reached. This represents a great risk to the minute organisms, especially Calanoida Copepoda, which act as the second level within the trophic chain. Thus, Calanoida, which are the most threatened organisms, face high predation danger; this may explain the decrease in calanoid numbers in such habitats and their increase in the open water. There have been several studies confirming this assumption.

Williams et al. [68] studied the food items of different coral reef fish and their stomach contents and indicated their feeding on endemic copepod species. Moreover, the reef inhabitants and zooplanktivores have intense predation roles on copepod species in the flowing water over the coral reefs [73–75]. Similarly, El-Serehy and Abdel-Rahman [62] suggested that Calanoida are considered the first preferable food item among Copepod orders for the planktivores in the coral reef. This may explain the absence of *Clausocalanus arcuicornis* from coral reef habitats during the warm season and *Temora stylifera* during the cold season.

All these above-mentioned species have been recorded in this study and could explain why they were not restricted to coral reef habitats but found among different habitats—they may be escaping from predators. Therefore, *Acrocalanus gibber* was restricted to the open deep-water habitat and not recorded at any other habitat, while it disappeared during the winter. *Clausocalanus arcuicornis* and *Paracalanus aculeatus* were recorded at three habitats during the cold and warm seasons (SSL, ODW, and CR). *Phaenna spinifera* was found in the coral reef and open deep water during the warm season, and disappeared during the cold season, while *Temora stylifera* was an inhabitant of the coral reef and open deep water during the cold season, and open deep water and sheltered habitats during the warm season. During the cold season, there were no species restricted to sheltered shallow habitats, and this may be owing to the effect of rainfall in changing the environmental conditions in these semi-enclosed areas, such as changes in water salinity, which can result in death, escape of organisms, or act as a barrier against invasion of organisms. These barriers may be ecological [76] or biological such as predators. In this study, Clausocalanidae were represented by three species, *C. arcuicornis*, *C. furcatus*, and *C. pergens*; *Clausocalanus* is widespread and well-known in aquatic environments worldwide. Cornils et al. [77] noted that this genus is one of the most abundant and common genera of Calanoida in ocean waters, especially in the tropical and subtropical regions, as it can tolerate severe environmental conditions, especially high temperature.

Among all recorded species, *Nannocalanus minor* was the only species that could exist in all habitats during the cold and warm seasons; this may be because of the special characteristics that this species possesses, including (unlike many calanoid copepods) continuous reproduction (up to 5 generations/year) through most of its range, resulting in multiple generations produced [78]. In addition, this species is a cosmopolitan recorded from different water bodies; the Pacific Ocean [79],

Atlantic Ocean [80], Indian Ocean [81], subtropical oceans [82], and Mediterranean Sea [45], which reflects its great ability to withstand various environmental conditions.

Raymont [83] noticed that the breeding positively correlated with the water temperature; hence, at the warm seas, most species reproduce all the year-round. It is common that immature forms of Copepoda (nauplius larvae and copepodites) reach their maximum abundance (that can rises to bloom) during summer as mentioned by several authors [2,84,85].

## 5. Conclusions

The Red Sea is an exquisite ecosystem with many diverse habitats. As the habitat changes, the environmental conditions and biodiversity differ. Copepoda, especially Calanoida, which represents the larger proportion of zooplankton, are second level producers in the marine food web. These minute organisms are affected by changes in these habitats, where some of them thrive and others disappear. Furthermore, the present study observed that some Calanoida species were widespread and adapted to exist in all habitats and seasons, such as *Nannocalanus minor*. This species has a great tolerance to live in different environmental conditions, making it a focal species that needs to be further studied. Understanding habitat preferences provides critical information for habitat management and/or native species conservation. The present study extends basic data regarding the Calanoida community and diversity in the different habitats of the Hurghada shelf, north-western Red Sea. One of the difficulties encountered in the current study is that it is the first, to the best of our knowledge, to compare the biodiversity and abundance of Calanoida copepods species in the various Red Sea habitats (coral reefs, shallow lagoons, seagrass, and open deep water). Owing to the importance of comparative studies of biodiversity and environmental conditions in different habitats of the Red Sea, as well as the lack of such studies, there is an urgent need to conduct more studies—especially on the communities of zooplankton and copepods—and expand the study of these habitats to include other habitats, such as mangroves and rocky habitats.

**Author Contributions:** H.A-T.: Conceptualization, data curation, formal analysis, investigation, methodology, visualization, writing—original draft, writing—review and editing, project administration. A.E.-S: Funding acquisition, resources, visualization, writing—original draft, writing—review and editing, project administration. A.A.: Funding acquisition, resources, visualization, writing—original draft, project administration. M.M.M.: Investigation, methodology, visualization, writing—original draft. M.A.: Data curation, formal analysis, investigation, methodology, visualization, writing—original draft, writing—review and editing. All authors have read and agreed to the published version of the manuscript.

**Funding:** This work was supported by the Vice Deanship of Research Chairs at King Saud University.

**Acknowledgments:** The authors wish to express their gratitude to the editor, the reviewers for their valuable comments and assistance in revising the manuscript. This project was financially supported by the Vice Deanship of Research Chairs at King Saud University.

**Conflicts of Interest:** The authors declare no conflict of interest.

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
