# Peer review of "Biodiversity of Calanoida Copepoda in Different Habitats of the North-Western Red Sea (Hurghada Shelf)"

_water, doi:10.3390/w12030656_

Round 1
Reviewer 1 Report
The manuscript is well written and clear. for this reason, probably, my attention has been addressed to minor aspects. Not only due to the politeness of the writing, however, I warmly suggest to avoid misunderstanding use of terms as density, stage, etc, and a correct use of the latin in taxa names (for example: Copepoda Calanoida and not copepods calanoids).
In any case all these my indications are directly reported on the pdf file.
Regards

Author Response
Response to Reviewer 1 Comments
Point 1: The manuscript is well written and clear. for this reason, probably, my attention has been addressed to minor aspects. Not only due to the politeness of the writing, however, I warmly suggest to avoid misunderstanding use of terms as density, stage, etc,
Response 1: We want to extend our sincere thanks and appreciation to the reviewer for his efforts as well as for his valuable comments. All the highlighted words: "density" along the manuscript has been changed to "abundance" as reviewer's comment.
Point 2: correct use of the Latin in taxa names (for example: Copepoda Calanoida and not copepods calanoids).
Response 2: All the highlighted words: "copepods" and "calanoids" along the manuscript have been corrected using of Latin in taxa names to be "Copepoda" and "Calanoida" as reviewer's comment.
Point 3: Line 44: change "water" to "marine water".
Response 3: "water" has been changed to "marine water".
Point 4: Line 44: change "vertical" to "daily vertical"
Response 4: Line 44: "vertical" has been changed to "daily vertical"
Point 5: Line 95: Figure 1, please do not use the red color for the geographic square.
Response 5: Figure 1: The red color has been changed to purple.
Point 6: Line 110: change "organisms" to "individuals".
Response 6: Line 110: "organisms" has been changed to "individuals".
Point 7: Line 118: It is important to clarify that only adults have been identified (is it true?). Somewhere else in the text it is reported that immature stages have been counted altogether. Please specify this point.
Response 7: The sentence "Only adults have been identified, while the immature stages have been counted altogether" has been added. (Lines 121:122).
Point 8: Line 156: please re-arrange the present sentence according the exclusive presence (not the absence). For example: "Six species were exclusive of the warm period, while four species was exclusive of the cold period".
Response 8: The sentence "Six species were not recorded during the cold season, while four species were not recorded during the warm season" has been rearranged as the reviewer suggestion, to: "Six species were exclusive of the warm period, while four species was exclusive of the cold period". (Line 156:157)
Point 9: Line 161: change the word "densities" to "abundances"
Response 9: The word "densities" has been changed to "abundances"
Point 10: Line 167: Table 2. Families are arranged, in the table, according a not immediately evident rule. Please adopt an evolutionary ranking (clear to only specialists), or an alphabetic order (better).
Response 10: Families have been arranged, in the table, according to alphabetic order.
Point 11: Table 2. Add "e" to the end of the species name
Response 11: Labidocera madura Scott A.1909 has been changed to Labidocera madurae Scott A.1909.
Point 12: Line 177: Correct "detruncate" to "detruncata"
Response 12: "detruncate" has been corrected to "detruncata".
Point 13: Line 193: "Labidocera" what species?
Response 13: Labidocera as a genus. It has been added in the text of the manuscript.
Point 14: Line 196: italics. , but I suggest Acartiidae (not italics) to be more precise
Response 14: "Acartia" has been changed to "Acartiidae" as the reviewer suggested
Point 15: Line 210: The reviewer highlighted the sentence "A. erythraea and M. clausi occurred with a value of 8% in one station during this season." and asked "Among the less abundant species ......?"
Response 15: The answer is yes, however, we don’t treat the abundance in this section, but we deal with species just from the point of occurrence (present or absent).
Point 16: Line 242, 243, and 244: The format of the species names "Acartia erythraea, Candacia simplex, Centropages violaceus, and Temora stylifera" should be in italic.
Response 16: The species names has been modified to be italic format.
Point 17: Equations 3 and 5 : Correct Individual/m3 to Individuals/m3
Response 17: Corrected
Point 18: Line 284 and 302: Figure 5 and 6. Reviewer comment: Please pay attention to the names of species L. madurae, and to the writing of nauplius larvae and Copepodites: these names (not taxa) have to be written with a small initial. Finally, stages are nauplius and copepodites. each stage contains several ages. Hence it is wrong to say "copepodites stages"
Response 18: The species name Labidocera madurae has been corrected in the two figures, as well as nauplius larvae and copepodites have been written with a small initial. The two figures have been changed by the correct ones.
Point 19: Line 351: delete "most" from the phrase "the most dominant"
Response 19: The "most" has been deleted.
Point 20: Line 352: the word "metazooplankton" is true! Or it is "mesozooplankton"?
Response 20: Yes, "metazooplankton" is the intended word, it is very common in description of groups such as Rotifera, Copepoda, larvaceans etc., and it is previously used in several manuscripts; such as:
"Jürgens & Jeppesen, 2000 (https://doi.org/10.1093/plankt/22.6.1047);
Jaspers et al., 2009 (https://doi.org/10.1093/plankt/fbp002);
Rekik et al., 2018 (https://doi.org/10.1007/s13201-018-0744-4)"
Point 21: Line 352: "Calanoids are relatively the most dominant among copepod orders". The reviewer asked to delete "the most".
Response 21: If we delete the word "most", the meaning will change to "Calanoida are relatively prevalent among the Copepoda orders". This is not intended, but what is meant is that this order is the most dominant at all.
Point 22: Line 360: delete "stages" from copepodite stages.Response 22: the word "stages" has been deleted.
Point 23: Line 388: Authors correctly refer to Copepoda and Calanoida, to stress that the paper 62 deals with Copepoda in general, and the present manuscript deals with Calanoida only. I suggest to operate homogeneous (and more clear) comparisons and using, from paper 62, only data for Calanoida.
Response 23: The paragraph was rewritten and calanoid numbers in the mentioned paper were included to make a homogeneous comparison. However, the numbers of copepods as a general were left to clarify that competition and environmental conditions in these areas affect copepod communities in general, including Calanoida, as the following (Lines: 406 : 412):
"[62] who observed higher decreases in the number of Copepoda species as general (total 48 species), specially Calanoida (35 species) in the coral reefs of the Red Sea at Hibika and Abu Galum than in the offshore sandy stations of Nuweiba and Ras Burka (74 Copepoda species involve 47 Calanoida), Where 26 species of Copepoda were absent, including 12 species of the order Calanoida alone. Moreover, in the current study, the highest number of calanoid species was recorded from open water habitats during the warm and cold seasons (26 and 31 species, respectively)."
Point 24: Line 399: The reviewer mentioned about Abo-Taleb and Gharib [2] that "also this paper deals with the whole Copepoda world. Please restrict the discussion to the only Calanoida.
Response 24: We have cited this paper in this paragraph not for talking about Copepoda or Calanoida, but to indicates that the seagrass environment is attractive for fish larvae, please, revise the phrase: "seagrass are the preferred habitats for the juveniles and fish larvae more than any other investigated habitat".
Point 25: Line 404: " Juveniles" small initial
Response 25: It has been corrected in (Line: 425)
Point 26: Line 405: "spp" not italics
Response 26: It has been corrected in (Line: 429)
Point 27: Line 419: Reviewer stated that "please pay attention to the use of this term endemic, in faunal studies, means existing only in one place".
Response 27: Williams et al., 1988 used, exactly, this term in his research paper more than once, and the scientific honesty necessitates transferring the term as it is. Also, many scientists use this term to describe this situation as in the current research. It is possible to refer to the research to confirm this "Cross-shelf distribution of copepods and fish larvae across the central Great Barrier Reef", (https://doi.org/10.1007/BF00392565)
Point 28: Line 419: "Clausocalaniodae was" reviewer stated that "Clausocalanidae (not -calanoidae) and no italics (it is a family name), being a family name it wants "were" and not "was".
Response 28: "Clausocalaniodae was" has been changed to "Clausocalanidae were"
We would like to extend our sincere thanks and appreciation to the reviewers and editorial board. In fact, their comments and guidance added a lot to the research and increased its scientific content. Therefore, the words cannot express their gratitude for their time and effort they put in evaluating this research.
Reviewer 2 Report
The present paper delivers basic information on density and species diversity of calanoid copepods from the Hurghada shelf. I believe biodiversity information is valuable from the regions, although it would be much better to include several biological, and physicochemical factors including nutrients, primary production values, rainfall, etc.
It would be better to use a plankton net with a larger mesh size, and the current plankton net seems to fit for net phytoplankton than zooplankton.
I guess that authors have a high rate of larval stages of calanoid copepods, nauplius, and copepodites. It would be better to show the difference in larval stages among the habitats, and also between the seasons.
Although the present paper is focusing on calanoid copepods, cyclopoid, and harpacticoid copepods can be important in CR, SSL, and SG areas. It would be better if the authors could discuss this matter as well before the publication.
Author Response
Response to Reviewer 2 Comments
Point 1: The present paper delivers basic information on density and species diversity of calanoid copepods from the Hurghada shelf. I believe biodiversity information is valuable from the regions, although it would be much better to include several biological, and physicochemical factors including nutrients, primary production values, rainfall, etc
Response 1: We want to extend our sincere thanks and appreciation to the reviewer for his efforts as well as for his comments. We also like to make it clear that this work was conducted based on biological data in addition to the environmental data that could be measured, as possible as we can, such as the water temperature, dissolved oxygen, pH values and water salinity. Certainly, more environmental measurements will have a major impact in explaining and clarifying these relationships, so we will make sure to do that in the next research work.
Point 2: It would be better to use a plankton net with a larger mesh size, and the current plankton net seems to fit for net phytoplankton than zooplankton.
Response 2: in some previous works, we used nets with 100 µm mesh size in collecting samples, but we faced many comments from reviewers stating that the immature stages may have a size of lesser than 100 microns and asked us to use nets have smaller holes size so we used a 55 µm network to can collect the smaller size forms, while our colleagues used nets with 25 µm mesh size In the collection of phytoplankton. However, the reviewer advice will be take in our consideration in the next work.
Point 3: I guess that authors have a high rate of larval stages of calanoid copepods, nauplius, and copepodites. It would be better to show the difference in larval stages among the habitats, and also between the seasons.
Response 3: To add the results of the difference in larval stages among the habitats and between the seasons, the paragraph (Lines: 240 to 249) and Table (5) have been added to the result section in the new submitted manuscript. Moreover, the paragraph (Lines: 474 to 477) has been added to the discussion part as well as the new references [83-85] (Lines: 724 to 730) have been added to the reference part.
Point 4: Although the present paper is focusing on calanoid copepods, cyclopoid, and harpacticoid copepods can be important in CR, SSL, and SG areas. It would be better if the authors could discuss this matter as well before the publication.
Response 4: There is no doubt that the remaining copepod orders (cyclopoid, and harpacticoid copepods, as well as Poecilostomatoida) are important in CR, SSL, and SG areas as the reviewer reported. However, the current manuscript is a very specific study to Clanoida Copepoda while the rest of the groups were discussed, in general, and their presence in different habitats in our previous work (Abo-Taleb and Gharib, 2018. doi:10.21608/ejabf.2018.9456).
We would like to extend our sincere thanks and appreciation to the reviewers and editorial board. In fact, their comments and guidance added a lot to the research and increased its scientific content. Therefore, the words cannot express their gratitude for their time and effort they put in evaluating this research.
Round 2
Reviewer 2 Report
Unfortunately, the authors were not able to improve all the points commented. However, I would understand the authors' difficulties to improve the manuscript since it is related to basic methods. As biodiversity information from the less surveyed area, it is certainly worth to be published.